# Mitigation Effect of Helmholtz Resonator on the Micro-Pressure Wave Amplitude of a 600-km/h Maglev Train Tunnel

Dian-Qian Li [1,2], Ming-Zhi Yang [1,2,*], Tong-Tong Lin [1,2], Sha Zhong [1,2] and Peng Yang [1,2]

1   Key Laboratory of Traffic Safety on Track, Ministry of Education, Central South University, Changsha 410075, China
2   School of Traffic & Transportation Engineering, Central South University, Changsha 410075, China
*   Correspondence: 205183@csu.edu.cn; Tel.: +86-13787088412

**Abstract:** A 600-km/h maglev train can effectively close the speed gap between civil aviation and rail-based trains, thereby alleviating the conflict between the existing demand and actual capacity. However, the hazards caused by the micro-pressure wave amplitude of the tunnel that occurs when the train is running at higher speeds are also unacceptable. At this stage, mitigation measures to control the amplitude of micro-pressure waves generated by maglev trains at 600 km/h within reasonable limits are urgent to develop new mitigation measures. In this study, a three-dimensional, compressible, unsteady SST K–ω equation turbulence model, and an overlapping grid technique were used to investigate the mechanism and mitigation effect of Helmholtz resonators with different arrangement schemes on the micro-pressure wave amplitude at a tunnel exit in conjunction with a 600-km/h maglev train dynamic model test. The results of the study showed that a pressure wave forms when the train enters the tunnel and passes through the Helmholtz resonator. This in turn leads to resonance of air column at its neck, which causes pressure wave energy dissipation as the incident wave frequency is in the resonator band. This suppresses the rise of the initial compressional wave gradient, resulting in an effective reduction in the micro-pressure wave amplitude at the tunnel exit. Compared to conventional tunnels, the Helmholtz resonator scheme with a 94-cavity new tunnel resulted in a 31.87% reduction in the micro-pressure wave amplitude at 20 m from the tunnel exit but a 16.69% increase in the maximum pressure at the tunnel wall. After the Helmholtz resonators were arranged according to the 72-cavity optimization scheme, the maximum pressure at the tunnel wall decreased by 10.57% when compared with that before optimization. However, the micro-pressure wave mitigation effect at 20 m from the tunnel exit did not significantly differ from that before the optimization.

**Keywords:** maglev trains; Helmholtz resonators; micro-pressure wave; moving model test; numerical simulation



## 1. Introduction

With the progress in science and technology, the safety and comfort of rail transportation and operating speeds have also improved. Currently, the highest commercial operating speed of wheeled rail high-speed trains is 350 km/h for CR series trains; however, it is difficult to further break the shackles of commercial operating speed owing to factors such as wheeled rail friction and economic efficiency. As a novel transport system, high-speed maglev trains exhibit the advantages of low total resistance, low noise, and more space for increasing speed. The maximum commercial operating speed of a special maglev line at the Shanghai Airport in China is 430 km/h, and the superconducting Central Shinkansen in Japan, which is expected to be completed in 2027, is designed to operate at 505 km/h. Given that in civil aviation, aircrafts generally fly at 800 km/h, the development of a 600-km/h maglev transport system to close the speed gap between high-speed trains and civil aviation has become an important development goal for global rail transport power.

Compared to conventional high-speed trains, the severity of problems due to aerodynamic effects during the operation of a 600-km/h maglev train will be more pronounced.

When a high-speed maglev train passes through a tunnel, an initial compression wave is formed in front of the vehicle, which propagates forward at the local speed of sound and reaches the exit of the tunnel, and part of it radiates outward in the form of a low-frequency pulse wave, forming a micro-pressure wave. In previous studies, it was shown that the amplitude of micro-pressure waves at the tunnel exit is proportional to the third power of the train's operating speed. As 600-km/h maglev trains pass through the tunnel, the tunnel entrance encounters the problem of high micro-pressure wave amplitude along with the sonic boom phenomenon, causing violent low-frequency vibrations of light structures, house doors, and windows around the entrance. This in turn affects the service life of nearby buildings and causes disturbance to residents [1]. In China, mountains account for two-thirds of the country's land area, and many long tunnels exist [2]. Furthermore, the number of tunnels, construction scale, and operational mileage are the highest in the world. Hence, the problem of excessive micro-pressure wave amplitudes is almost unavoidable during the operation of 600-km/h maglev trains.

Since the discovery of the micro-pressure wave problem on the Shinkansen in Japan in 1975 [3], many studies have been conducted and breakthroughs on the phenomenon of excessive micro-pressure waves have been reported in the field of high-speed trains. Yamamoto et al. [4] examined the formation process and mitigation measures of micro-pressure waves via model tests and field measurements and proposed a calculation method and three-dimensional angle model for micro-pressure waves at tunnel openings. Tebbutt et al. [5] initially proposed the arrangement of Helmholtz resonators in tunnels and optimized their dimensions using a genetic algorithm. Then, they applied a quasi-one-dimensional equation to simulate the propagation of acoustic waves in tunnels. By comparing the before and after results, they proved that Helmholtz resonators in tunnels of different lengths can effectively reduce the micro-pressure wave amplitude. Zhang et al. [6,7] observed in their study that the initial increase in pressure at all measurement points at the tunnel exit was mainly determined by the initial compression wave generated at the time of train entry. Using a 1:20 dynamic model test rig, they measured the transient pressure and micro-pressure waves of a 350-km/h train model passing through various tunnel models and observed that the combination of a buffer structure with a roof hole and a cap-shaped diagonal cut tunnel opening can effectively reduce the micro-pressure wave amplitude. Kim et al. [8] combined bionic technology with a buffer structure at a tunnel entrance and determined that a buffer structure with air slits can effectively reduce the micro-pressure wave amplitude. Furthermore, they developed an analytical model for predicting micro-pressure waves. Wang et al. [9] conducted a comparative study on a variety of buffer structures at 400-km/h high-speed railway tunnel entrances. They proposed to set up an open-hole slant-cut equal-section enlarged section buffer structure of a certain length at the tunnel entrance such that the micro-pressure wave amplitude at tunnel entrances of 5 km and below could satisfy the national regulation standard.

In recent years, with the increase in train speed demand, researchers have focused on the phenomenon of high micro-pressure wave amplitude in 600-km/h maglev trains. Howe et al. [10] proposed continuum theory to examine the effect of maglev trains entering tunnels at high Mach numbers on micro-pressure waves. Jia et al. [11] examined the distribution pattern of peak pressure in the tunnel during the rendezvous of 600-km/h maglev trains and concluded that the railway tunnel structure did not satisfy the requirements of high-speed maglev trains passing through the tunnel, and pressure-reducing measures, such as increasing the clear space area of the tunnel or installing additional shafts, should be considered. Lin et al. [12] used the overlapping grid method to investigate the initial compression waves and micro-pressure waves when traversing a tunnel at the head of a 600-km/h high-speed maglev train using three different arch structures: single-arch, double-arch, and triple-arch. They observed that an increase in the arch structure of the maglev train can effectively reduce the initial compression wave gradient in the tunnel and micro-pressure wave amplitude at the tunnel exit. Zhang et al. [13] proposed a buffer structure with an arched lattice design that can further dissipate the energy of micro-pressure

waves through a semi-enclosed decompression region formed by the buffer structure. Niu et al. [14] examined the effect of gas deflectors on the aerodynamic effects generated by high-speed maglev trains passing through tunnels at a speed of 600 km/h. They observed that although the deflectors can effectively suppress the pressure between the tunnel wall and train surface, they can only provide limited relief with respect to the micro-pressure wave amplitude. Yang et al. [15] designed and developed the world's only dynamic model test platform for speeds up to 680 km/h. This in turn significantly reduced the cost for testing maglev trains at 600 km/h and improved the accuracy of the test data. Han et al. [16] used numerical simulations to investigate the aerodynamic effects of maglev trains passing through tunnels at different speeds and observed that the power exponential relationship between the speed of the maglev train and amplitude of the micro-pressure waves increases as the speed of the train increases. Furthermore, they observed sub-reflection waves.

The Helmholtz resonator is a chamber installed inside a tunnel comprising a cavity and corresponding neck passage. The pressure wave generated by the train passing at high speed forms a column of air with a certain mass in its neck, which can be considered as a spring oscillator. The air inside the chamber can be considered as a spring, and the neck and chamber form an elastic vibrating system, which resonates when the acoustic frequency of the airflow is similar to the intrinsic frequency of the vibrating system [17]. When a vibrating system resonates, most of its energy dissipates. Each Helmholtz resonator has its own resonant frequency, and Xu et al. [18] and Cai et al. [19] observed that the resonant band of a single-cavity, double-necked Helmholtz resonator system is wider than that of a conventional single-cavity, single-necked Helmholtz resonator. Therefore, all the Helmholtz resonators described in this study are single-cavity, double-hole-type Helmholtz resonators.

In summary, existing mitigation measures cannot control the micro-pressure wave amplitude at the tunnel exit within a reasonable range for maglev train systems with speeds of 600 km/h and higher. Furthermore, the mitigation mechanism and specific arrangement of Helmholtz resonators for micro-pressure waves in high-speed train tunnels are not yet fully understood. Therefore, in this study, an aerodynamic model test system of a 600-km/h maglev train, at the Key Laboratory of the Ministry of Rail Transportation Safety of Central South University, was used to combine moving model tests and numerical simulations to determine a layout solution with an excellent mitigation effect on the micro-pressure wave amplitude.

## 2. Numerical Model

### 2.1. Maglev Train and Tunnel Model

In this study, a 600-km/h magnetic levitation test train was developed in China as the object of study. The train model was scaled down to 1:20 prior to the numerical simulation because the full y+ wall treatment was used and the y+ needed to be controlled to be less than 1 or greater than 30. Figure 1 illustrates the training model. At full scale, the maglev train has a height of h = 4.14 m, cross-sectional area of 11.95 m$^2$, T-shaped track beam with a track surface of 1.09 m from the ground, and a bottom surface at 0.3 m from the ground. Using the maglev train height as a benchmark for data in this study for dimensionless processing, the car length is 20.1 h, car width is 0.89 h, head car is 7.04 h, middle car is 5.97 h, and nose height is 0.34 h. The entire car is composed of four parts: frame, wheel assembly, outer shell of the rolling stock, and an on-board test system [20].

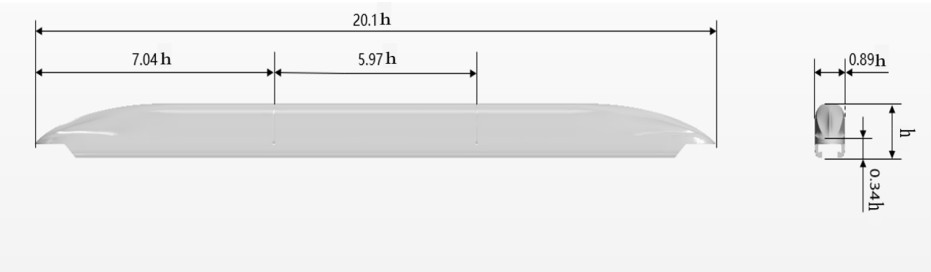

**Figure 1.** Maglev train model.

Owing to the extremely strict requirements of the equipment form factor at the tunnel limit, the arrangement of the equipment inside the tunnel should be compact. Therefore, it is particularly important to reduce the neck length and influence the flow field while expanding the spectral range of the resonator [21]. Su et al. [22] examined that the Helmholtz resonator neck can be extended inward to reduce the resonant frequency while maintaining its surface finish.

In summary, all the Helmholtz resonators in this study were single-cavity double-necked, with an inward extending neck, a neck hole diameter length of d = 0.14 h, and a neck channel depth of l = 0.1 h (as shown in Figure 2). In the experiments in this study, a number of different Helmholtz resonator arrangements with 94, 72, and 64 cavities were set up as follows:

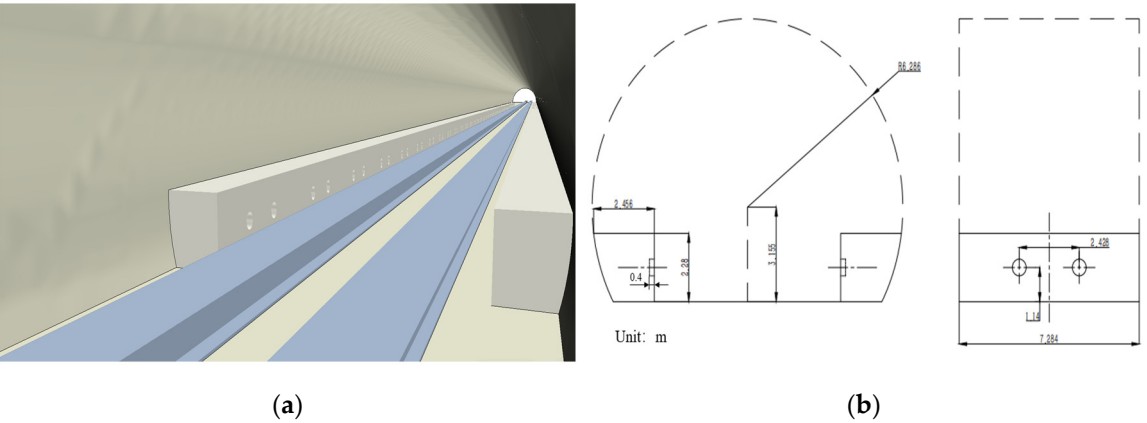

(**a**)

(**b**)

**Figure 2.** Helmholtz resonator. (**a**) Helmholtz resonators 3D simulation; (**b**) Helmholtz resonator structure.

1. Conventional tunnel without Helmholtz resonators and without any additional structures.
2. A 94-cavity new tunnel comprising 94 cavities arranged continuously without intervals on both sides of the tunnel, starting at 4.78 h from the entrance and ending at 6.83 h from the exit of the tunnel.
3. A 72-cavity new tunnel 1, the 72 cavities were divided into 24 groups, each group consisting of 3 cavities, with an interval of 1.76 h between each group, starting inside the tunnel at 4.78 h from the entrance and ending at 6.83 h from the tunnel exit.
4. A 64-cavity new tunnel, the 64 cavities were divided into 32 groups, each group consisting of 2 cavities, with an interval of 1.76 h between each group, starting inside the tunnel at 2.39 h from the entrance and ending at 6.83 h from the tunnel exit.
5. A 72-cavity new tunnel 2, the 72 cavities were divided into 24 groups, each group consisting of 3 cavities, with an interval of 1 h between each group, arranged from 2.39 h inside the tunnel from the entrance and ending at 15.18 h from the exit of the tunnel. The specific programmer is listed in Table 1.

**Table 1.** Helmholtz resonator module arrangement.

|  | Conventional Tunnel | 94-Cavity New Tunnel | 72-Cavity New Tunnel 1 | 64-Cavity New Tunnel | 72-Cavity New Tunnel 2 |
|---|---|---|---|---|---|
| Number of Cavities | 0 | 94 | 72 | 64 | 72 |
| Distance from entrance | 0 | 4.78 h | 4.78 h | 2.39 h | 2.39 h |
| Distance to exit | 0 | 6.83 h | 6.83 h | 6.83 h | 15.18 h |
| Distance between two cavities | 0 | 0 | 1.76 h | 1.76 h | 1 h |
| Layout location | 0 | Bottom of both sides | Bottom of both sides | Bottom of both sides | Bottom of both sides |
| Layout options | 0 | Continuous layout | Arranged in groups at intervals (groups of 3 cavities) | Arranged in groups at intervals (2 chamber groups) | Arranged in groups at intervals (groups of 3 cavities) |

When multiple Helmholtz resonators are used, the distance between the necks of the chambers must be considered. When the distance between the different cavity necks is

too small or the cavity itself is too small, the effect of the Helmholtz resonators will be limited by each other. This reduces the relief effect as opposed to increasing it. Furthermore, Tebbutt et al. [5] introduced the formula for determining the interference between Helmholtz resonators as follows:

$$\kappa = \frac{V}{S_t \chi} \ll 1 \tag{1}$$

where $V$ denotes the volume of the Helmholtz resonator, $\chi$ denotes the hole spacing between the two resonators, and $S_t$ denotes the tunnel cross-sectional area.

Given that the Helmholtz resonator used in this study is a double hole, the hole spacing has more than one value. This calculation is considered according to the most unfavorable conditions, taking the minimum value of 5.055 m. Furthermore, $\kappa = 0.136$ is calculated to satisfy the conditions.

### 2.2. Calculation Domain and Measurement Point Layout

In this study, the overlapping mesh technique [20] was used to numerically simulate the relative motion between the tunnel and train. The size of the overlapping region was 43.48 h × 1.2 h × 1.2 h, consisting of the maglev train and track, and the stationary region consisted of the ground, tunnel, and external computational domain. The overlapping region moves along the tunnel at the operating speed of the train, and information is exchanged between the two via the overlapping grid exchange surface [16]. A two-lane tunnel with a cross-sectional area of 100 m² and tunnel length of 96.62 h, which has been widely used in engineering, was selected and applied to the test with a 1:20 scaling ratio.

In the numerical model, a calculation domain of 72.46 h in length, 14.49 h in height, and 28.99 h in width was set-up at both ends of the tunnel to ensure the stability of the tunnel entrance and exit flow fields and accuracy of the numerical simulation results. The car started at 24.15 h from the tunnel entrance and entered the tunnel at a speed of 166.6667 m/s. The head of the car started to enter the tunnel at the 0.03-s mark. Figure 3 shows the calculation model.

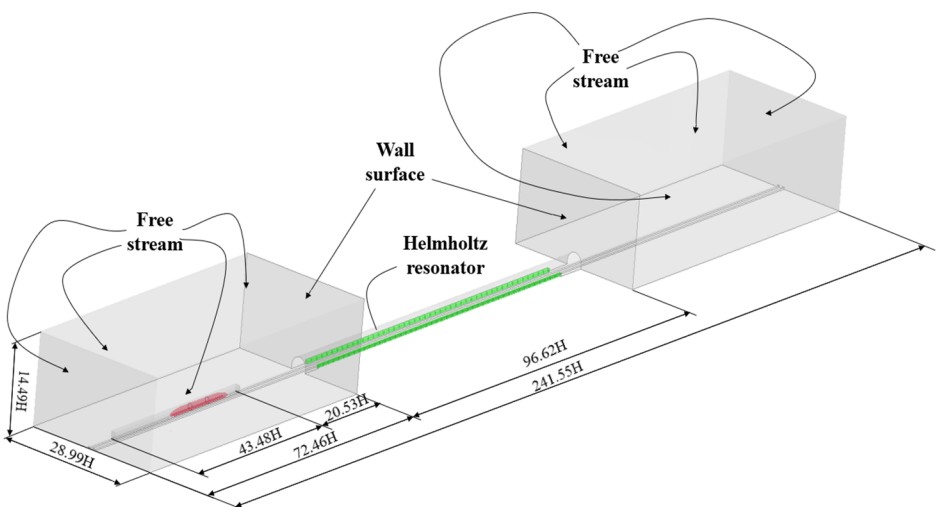

**Figure 3.** Computational Domain.

The pressure measurement points were mainly placed on the tunnel wall and at the tunnel exit to monitor the pressure fluctuations on the tunnel wall, initial compressional wave changes in the tunnel, and micro-pressure wave changes at the tunnel exit. The points are placed at the tunnel and the tunnel exit at a scale of 1:20, with the central point at the bottom of the section at the tunnel entrance as the origin, as shown in Figure 4. Furthermore, M-1, M-2, and M-3 were placed at 0.5 m, 1 m, and 2.5 m, respectively, from the exit of the model tunnel, corresponding to the realistic locations of 10 m, 20 m, and 50 m, respectively, from the tunnel exit.

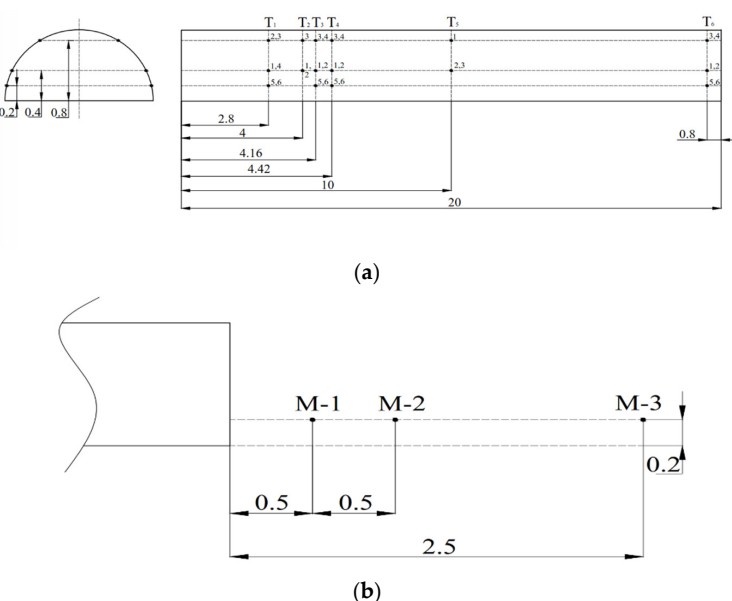

(a)

(b)

**Figure 4.** Location of measurement points. (**a**) Location of measurement points in the tunnel. (**b**) Layout of measurement points outside the tunnel.

*2.3. Grid Division Strategy*

The grid cells used for the numerical simulations were generated using STARCCM+ software, and the grid type was a cut-body grid. Adaptive encryption was used for the train and track surfaces to provide the mesh with an appropriate encryption size for the model area. The minimum mesh size for the train surface was $5.42 \times 10^{-3}$ h and that for the track surface was $2.72 \times 10^{-3}$ h. To simulate the development of the flow field on the train surface, a 20-layer boundary layer mesh with a growth rate of 1.2 and total thickness of 6 mm was applied to the surface of the maglev train. According to Iliadis' study [23], the tunnel surface and ground boundary layer slightly affect the propagation and diffusion of pressure waves in the tunnel. Therefore, the near-wall attached boundary layer grid is not used at the tunnel wall and ground. The wall treatment method corresponds to full y+ wall treatment, with a total grid volume of approximately 50,249,100 and grid volume of approximately 15 million in the overlapping grid area around the car body, as shown in Figure 5.

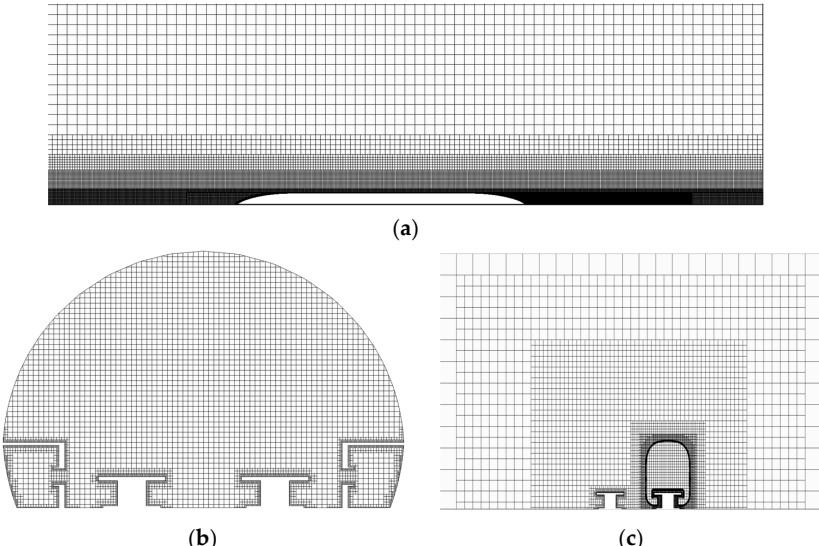

(a)

(b)                                    (c)

**Figure 5.** Grid generation details. (**a**) Longitudinal section of train; (**b**) Internal tunnel cross section; (**c**) Cross section of train.

The calculations were performed using the SST K–ω separated flow simulation based on the DDES turbulence simulation method, with the flow term set in mixed-BCD format while using the flow field equation of state for an ideal gas and separating the fluid temperatures. The boundary conditions in the computational domain were set to free-stream, incoming flow conditions were set to Mach 0, time step was set to $5.3 \times 10^{-5}$ s, and the second order time discrete format was selected. Furthermore, the maximum physical time and number of internal iterations were set to 0.3 s and 25, respectively.

### 2.4. Numerical Validation

2.4.1. Grid Independence Verification

To ensure that the numerical simulation results vary with the amount of mesh within a reasonable range and to avoid discrepancies in the numerical simulation owing to different mesh densities, in this study, we used the measurement data of M-2 at the exit of the model tunnel and verified the mesh irrelevance of the 600-km/h maglev train using three sets of meshes with different mesh densities. The mesh densities are as follows: coarse mesh accuracy with 32,667,800 meshes; medium mesh accuracy with 50,249,100 meshes; and fine mesh accuracy with 82,092,700 meshes, as shown in Figure 6. The results are shown in Figure 6.

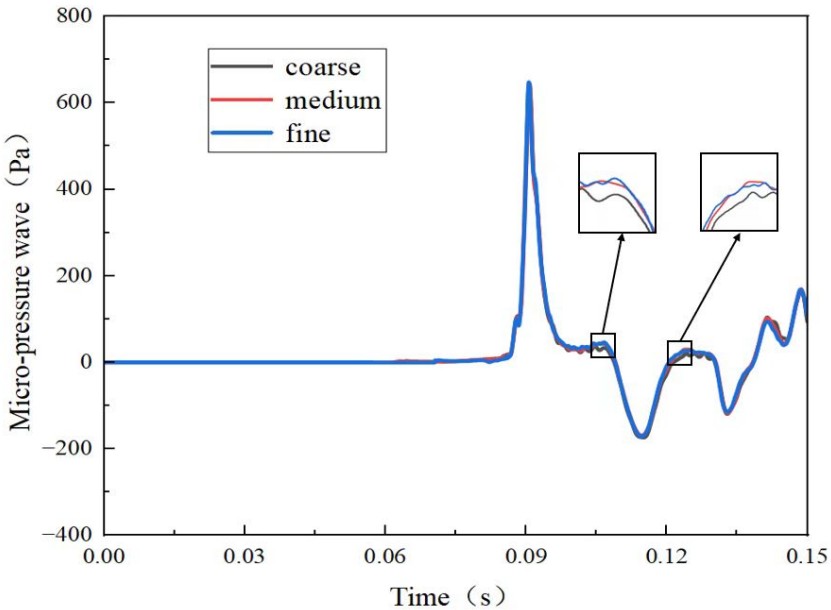

**Figure 6.** Micro-pressure waves at the tunnel exit for three grid densities with respect to time.

The surface of the comparison results shows that there are slight differences between the different grid volumes for the pressure–time profiles measured at the measurement points. However, the differences between the medium and fine grids are smaller when compared with the coarse grid, which has a lower demand for computational resources. Therefore, in this study, a medium grid was selected as the computational grid.

2.4.2. Moving Model Test Verification

To verify the accuracy of the numerical simulation results, the pressure–time curves obtained from the numerical simulation measurements were compared with the moving model test results using the conventional tunnel in the programmer table as an example. To ensure consistency between the experimental and computational boundary conditions, slip-ground boundary conditions were used for the numerical simulations. Moving model tests were conducted on the newly built 600-km/h high-speed maglev train moving model test platform at the High-Speed Train Research Center of Central South University. This is the world's largest test platform and the only "train pneumatic moving model test system", with CMA and CNAS (certificate number: CNAS L10220) qualifications [22–24].

In the moving model tests, the maglev train and tunnel models were scaled down to 1:20. Therefore, in the numerical simulations, the same scaled-down model was used for the maglev train and tunnel to ensure maximum consistency between the numerical simulation test setup and moving model test setup. Due to the rapid changes in test pressure of a moving model locomotive, it is necessary to ensure that the sensing and measurement channels have sufficient response speed. The sensor signal was amplified in two stages and filtered by second-order low-frequency filtering to eliminate high-frequency spurious interference. The entire range output was fed into a high-speed A/D converter for acquisition as a standard voltage signal of 0–5 V. Each pressure signal channel has an independent circuit structure and A/D converter, and synchronous sampling is controlled by the same time-based signal, thereby ensuring fast data acquisition and consistency in time and space for each channel. Figure 7 shows the scenario of the dynamic model for measuring pressure experimentally.

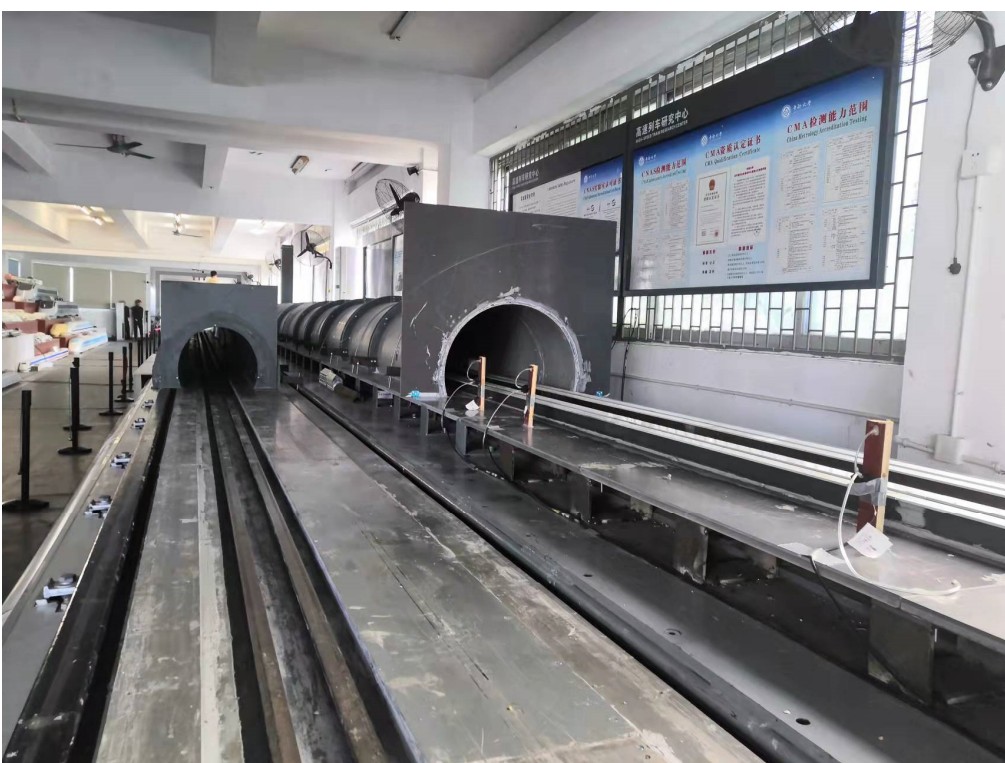

**Figure 7.** Moving model test site.

The results of this comparison are shown in Figure 8. As the train enters the tunnel, a compression wave forms in front of the train and propagates forward at the speed of sound. As the train reached the measurement point, the pressure–time curve appeared to be extremely high and then rapidly decreased. Subsequently, a micro-pressure wave was detected at the exit of the tunnel. A comparison of the numerical simulation results with the time course curves from the moving model experiments shows that there is a slight difference in either the trend or value. In terms of the pressure values at the tunnel exit, the maximum difference between the dynamic model test data and numerical simulation results is 6.93%, as listed in Table 2, which is in line with the actual engineering requirements. This indicates a high degree of confidence of the numerical simulation results.

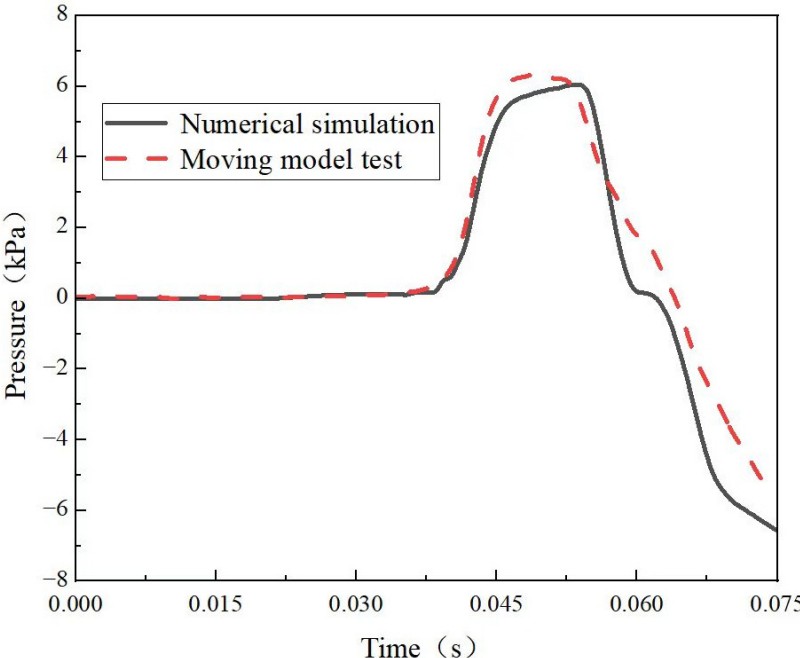

**Figure 8.** Comparison of dynamic model test and numerical simulation time travel curve.

**Table 2.** Comparison of moving model test and numerical simulation results.

| Tunnel Exit Pressure (Pa) | M-1 | M-2 | M-3 |
|---|---|---|---|
| Moving model test | 1070.556 | 681.97 | 313.9677 |
| Numerical simulation calculations | 1021.85 | 641.26 | 292.22 |
| Error | 4.55% | 5.97% | 6.93% |

This is because the results of the dynamic model test were derived using a number of sensors which were successively measured and then fitted to the data by the computer. Excessive fluctuations in ambient pressure or long measurement times affect the consistency and accuracy of the computer-generated time curves.

## 3. Results and Discussion

### 3.1. Three-Dimensional Effect Analysis of Tunnel

Figure 9 shows the relationship between the instantaneous pressure on the tunnel wall and propagation of the pressure wave when the maglev train passes through the tunnel at 600 km/h. In this figure, H denotes the train head displacement, W denotes the train tail displacement, C denotes the compression wave, and E denotes the expansion wave. Subscript H denotes the wave due to the evolution of the compressional wave induced by the entry of the train head, and W denotes the wave due to the evolution of the expansion wave induced by the entry of train tail, with the subscripts increasing when the wave is reflected. T6-1 to T6-6 are six measurement points in the same tunnel section that are arranged at different positions around the tunnel wall.

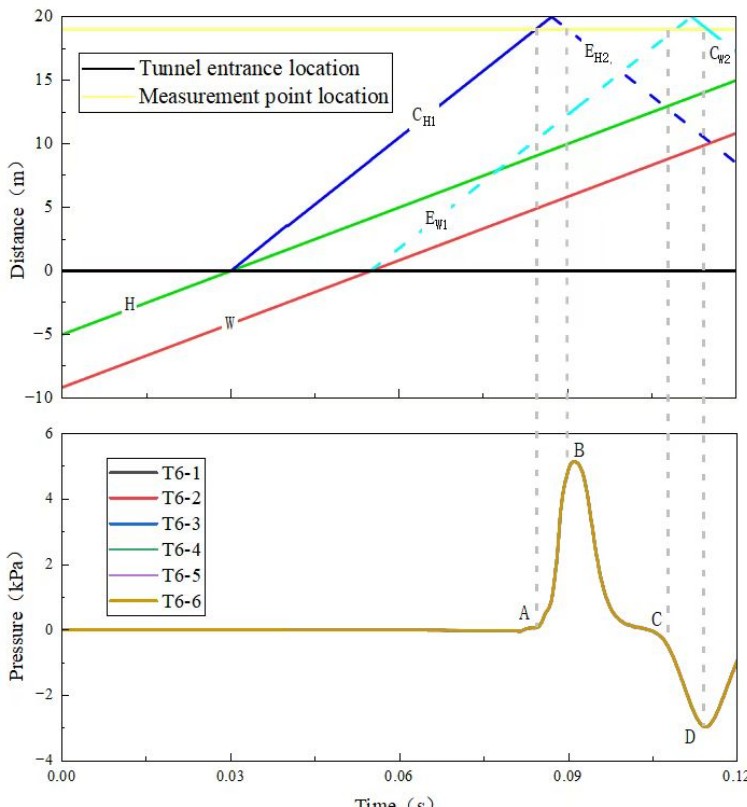

**Figure 9.** Pressure–time history curve of different measuring points in the same section.

At 0.03 s, the head of the train enters the tunnel. The air in front of the train is compressed, leading to the generation of the compression wave $C_{H1}$ and forward propagation at the speed of sound, reaching the measurement point at 0.0848 s, and resulting in an increase in pressure at A. The compression wave then reaches the tunnel entrance, and the expansion wave $E_{H2}$ is generated and arrives at the measurement point at 0.0894 s, resulting in a decrease in pressure at B. At 0.1097 s, the expansion wave $E_{W1}$ reaches the measurement point, and the measured pressure decreases again. When the expansion wave $E_{W1}$ reaches the tunnel exit, it generates a compression wave $T_{W2}$, which propagates towards the tunnel entrance, causing a pressure rise at D. During this process, the pressure profiles monitored at the different measurement points T6-1 to T6-6 overlap almost exactly, which is consistent with the results obtained by Liu et al. [24] for a 350-km/h train. This indicates that significant one-dimensional effects still dominate the tunnel when the model used in this study is operated at 600 km/h. Therefore, in subsequent studies, it is sufficient to set up only one monitoring point for different sections, and discussion of the magnitude of the effect of different measurement points on the test results in the same section is not required.

### 3.2. Analysis of the Effect of Helmholtz Resonators on Micro-Pressure Waves

3.2.1. Analysis of Helmholtz Resonator Resonance Conditions

When a high-speed maglev train enters a tunnel, similar to a piston entering a cylinder, the airflow is restricted by the tunnel walls, and the air at rest at the front of the train is violently compressed, resulting in a sudden increase in air pressure and formation of a compression wave that propagates forward along the tunnel at the speed of sound. The initial compression wave data measured in the numerical simulations are subjected to a dot-time Fourier transform to analyze its dominant frequency. The results of the Fourier transform are shown in Figure 10.

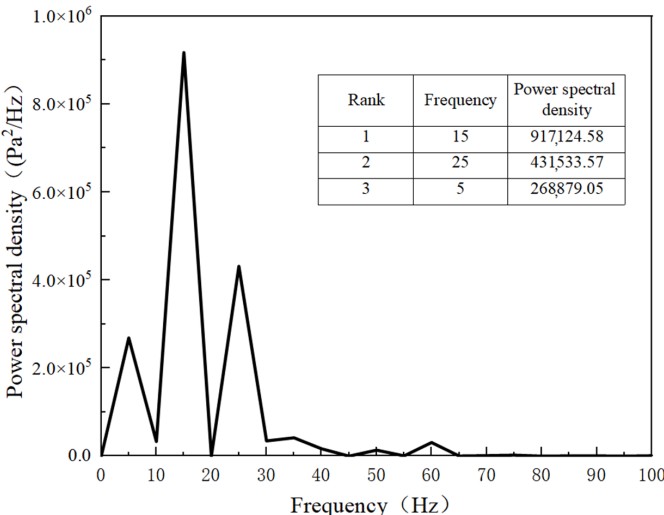

**Figure 10.** Main frequency distribution of the initial compressional wave.

As shown in the diagram, there are three main frequencies of the initial compressional wave: 5, 15, and 25 Hz. Specifically, 15 Hz corresponds to the main frequency carrying the highest energy. When the main frequency of the compressional wave is in the Helmholtz resonator band, resonance occurs. The resonance phenomenon consumes a large amount of energy and reduces the compressional wave gradient, thereby relieving the micro-pressure wave amplitude. According to Langfeldt [25] and Wang et al. [26], the resonant frequency of a two-cavity Helmholtz resonator can be derived using Equation (2) as follows:

$$f_0 = \frac{c}{2\pi} \sqrt{\frac{\rho}{V} \sum_{i=1}^{2} \frac{S_i}{M_i}} \qquad (2)$$

where $f_0$ denotes the resonant frequency of the Helmholtz resonator, $c$ denotes the speed of sound, $\rho$ denotes the density of air, $V$ denotes the cavity volume, $S_i$ denotes the cross-sectional area of the neck channel, and $M_i$ denotes the equivalent air mass (0.12).

The resonant frequency of the Helmholtz resonator is calculated as 14.63 Hz. As the incident wave frequency is in the resonant band of the Helmholtz resonator, it is necessary to calculate the resonant bandwidth. Before calculating the bandwidth, the neck aperture should be judged appropriately as follows:

$$\frac{0.01}{\sqrt{f_0}} < r < \frac{10}{f_0} \qquad (3)$$

where r denotes the radius of the neck channel of the Helmholtz resonator, and the size can be calculated to satisfy the moderate condition.

According to Wu [27] et al. and Herrero-Durá I [28], the bandwidth calculation method is shown by the Helmholtz resonator acoustic resistance, which is first calculated using Equation (4).

$$R_a = \frac{\sqrt{2\rho\eta\omega}}{\pi r^2} \left( \frac{l}{r} + 2 \right) \qquad (4)$$

where η denotes the shear viscosity coefficient of air, which is considered as $1.86 \times 10^{-5}$, and $\omega$ denotes the angular frequency, which can be calculated as: $\omega = 2\pi f_0$. Furthermore, l denotes the neck channel length of the Helmholtz resonator.

Combined with the theoretical study by Du et al. [29], Equation (5) is applied to calculate the acoustic resistivity ratio, and Equation (6) is applied to calculate the quality factor of the sound absorption structure of the Helmholtz resonator as follows:

$$X_S = \frac{R_a S}{\rho c} \tag{5}$$

$$Q_R = \frac{c}{2\pi(1 + X_S)D f_0} \tag{6}$$

where $X_S$ denotes the acoustic resistivity ratio, $Q_R$ denotes the quality factor, $D$ denotes the rigid length of the Helmholtz resonator cavity (i.e., the distance from the cavity mouth to the bottom of the cavity), and S denotes the sum of the cross-sectional areas of the multiple neck channels.

Finally, the bandwidth is calculated using Equation (7) as follows:

$$z = \sqrt{\left(\frac{1}{Q_R}\right)^2 + 1} \pm \frac{1}{2Q_R} \tag{7}$$

where $z$ denotes the range of frequency bands in the central resonant frequency.

The width of the resonant band is calculated between 13.76 Hz and 16.13 Hz. The initial compressional wave has three main frequencies. Among them, 15 Hz carries the highest energy and is in the resonant band, triggering the resonance of the Helmholtz resonator and causing a large amount of dissipation of the initial compressional wave energy.

### 3.2.2. Analysis of Micro-Pressure Wave AMPLITUDE Reduction Effect

The Helmholtz resonator is observed to be effective in reducing the micro-pressure wave amplitude. This is determined by comparing the micro-pressure wave amplitude data measured at measurement points M-1, M-2, and M-3 in the conventional tunnel and new 94-cavity tunnel. According to the CEN European Standard [30], the values measured at 20 m and 50 m at the exit of the tunnel are the micro-pressure wave amplitude values (i.e., at measurement points M-2 and M-3). As shown in Figure 11 and Table 3, the new 94-cavity tunnel with Helmholtz resonators shows a 31.87% and 33.23% decrease in the micro-pressure wave amplitude at M-2 and M-3, respectively, when compared with the numerical simulation results of the conventional tunneling scheme. By comparing the initial compressional gradient curves (Figure 12), it can be observed that the initial compressional gradient in the new 94-cavity tunnel exhibits a trend similar to that of the conventional tunnel, but with a 7.06% reduction in amplitude.

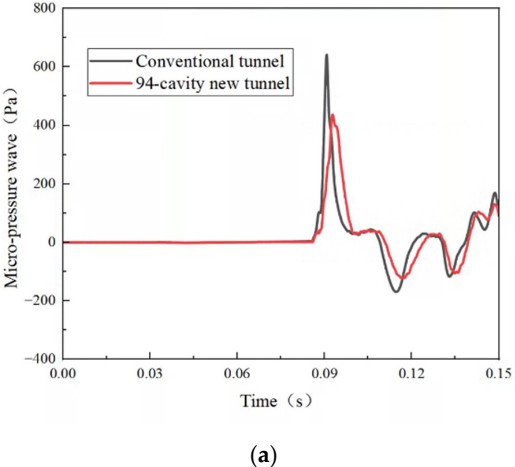
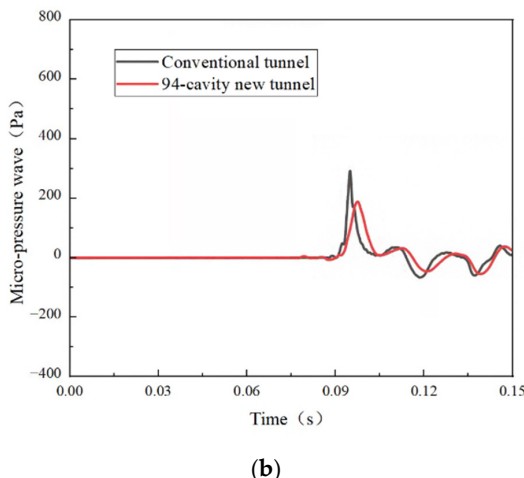

(**a**)                    (**b**)

**Figure 11.** Effect of different solutions on micro-pressure waves. (**a**) M-2; (**b**) M-3.

**Table 3.** Comparison of results of different solutions.

|  | M-1 | M-2 | M-3 | Wall Pressure in the Tunnel |
|---|---|---|---|---|
| Conventional tunnel | 1021.85 | 641.26 | 292.22 | 6149.85 |
| 94-Cavity New Tunnel | 703.87 | 436.91 | 195.12 | 7176.44 |
| Ratio | −31.12% | −31.87% | −33.23% | +16.69% |

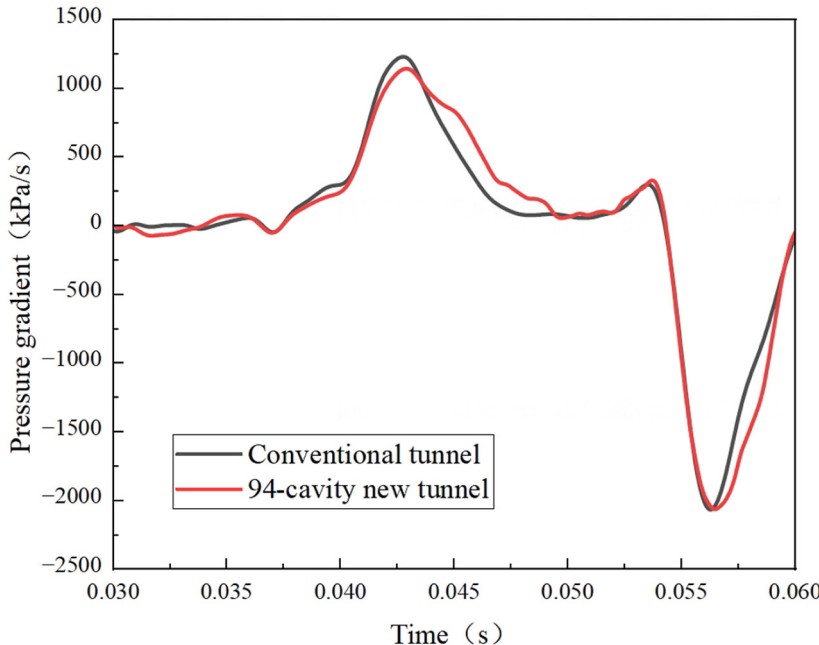

**Figure 12.** Initial compressional wave gradient comparison chart.

However, by comparing the values measured at the tunnel wall measurement points in the conventional tunnel with those in the new 94-cavity tunnel, it can be observed that the maximum tunnel wall pressure increases from 6149.85 Pa to 7176.44 Pa, an increase of 16.69%. This indicates that the presence of the Helmholtz resonator leads to an increase in pressure at the tunnel wall. The root cause of this phenomenon is the increased blockage ratio of the tunnel section owing to the presence of Helmholtz resonators [31,32], which increase the blockage ratio of the new 94-cavity tunnel by 18.9% when compared to that of the original tunnel. This in turn results in an increase in pressure.

### 3.3. Analysis of an Optimized Arrangement of Helmholtz Resonators

3.3.1. Optimized Solutions for Variable Cavities

In a previous study, Lu et al. [33–35] analyzed high-speed trains meeting in partially expanded sections of tunnels of different lengths and measured the pressure changes generated during the process. They concluded that the presence of partially expanded sections can effectively mitigate changes in the wall pressure of tunnel. Hence, a new type of tunnel with varying chambers, 72 and 64, was proposed to address the phenomenon of increasing wall pressure in the aforementioned tunnel.

The core of the new tunneling solution is based on the creation of multiple artificially locally expanded sections inside the tunnel by regularly reducing the Helmholtz resonators to sacrifice a small part of the micro-pressure wave mitigation effect in exchange for the tunnel walls, i.e., pressure fluctuations should be within reasonable limits. The specific arrangements and effects are presented in Table 1 and Figure 13.

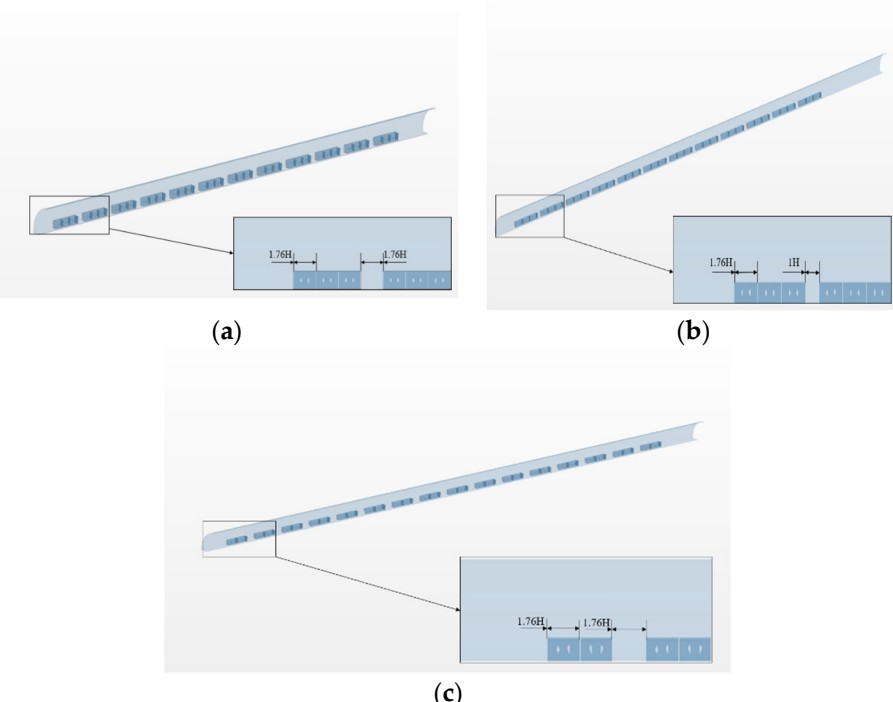

**Figure 13.** (**a**) 72-cavity new tunnel arrangement 1; (**b**) 72-cavity new tunnel arrangement 2; (**c**) 64-cavity new tunnel arrangement.

3.3.2. Analysis of the Effect of the Optimized Solution on the Pressure at the Tunnel Wall

The high-speed maglev train enters the tunnel at 0.03 s and an initial compression wave is generated at the front of the vehicle and propagates forward at the local speed of sound. The compression wave reaches the measurement point at 0.042 s, causing an initial increase in the pressure measured at the tunnel wall measurement point. At 0.055 s, the head car passes the measurement point, which decreases the measured pressure, whereas the tail car enters the tunnel and generates an expansion wave, which propagates forward at the speed of sound and reaches the measurement point at 0.0668 s. This in turn leads to a second drop in wall pressure values. At 0.0799 s, the tail car passes the measurement point, and thereby, the downward trend of the wall pressure curve ends. The measured value starts to rise until it returns to its initial state and stabilizes after a small oscillation.

Monitoring and comparison of conventional and various new tunnel wall pressures (as shown in Table 4 and Figure 14) reveal that the new tunnel solution can effectively mitigate the increasing trend in tunnel wall pressure values. The best relief is realized in the 72-cavity new tunnel 1, where the wall pressure increases by only 4.36% when compared to the original tunnel. A comparison of the rise rates of the 64-cavity new tunnel and new 72-cavity tunnel 2 shows that the narrowing of the distance between the two cavities leads to an increase in the tunnel wall pressure and different combinations of cavities affect the tunnel wall pressure.

**Table 4.** Comparison of pressure variations at the tunnel wall for different scenarios.

| Conventional Tunnel | 94-Cavity New Tunnel | 72-Cavity New Tunnel 2 | 64-Cavity New Tunnel | 72-Cavity New Tunnel 2 |
|---|---|---|---|---|
| 6149.85 Pa | 7176.44 Pa +16.69% | 6417.83 Pa +4.36% | 6655.52 Pa +8.22% | 6779.94 Pa +10.24% |

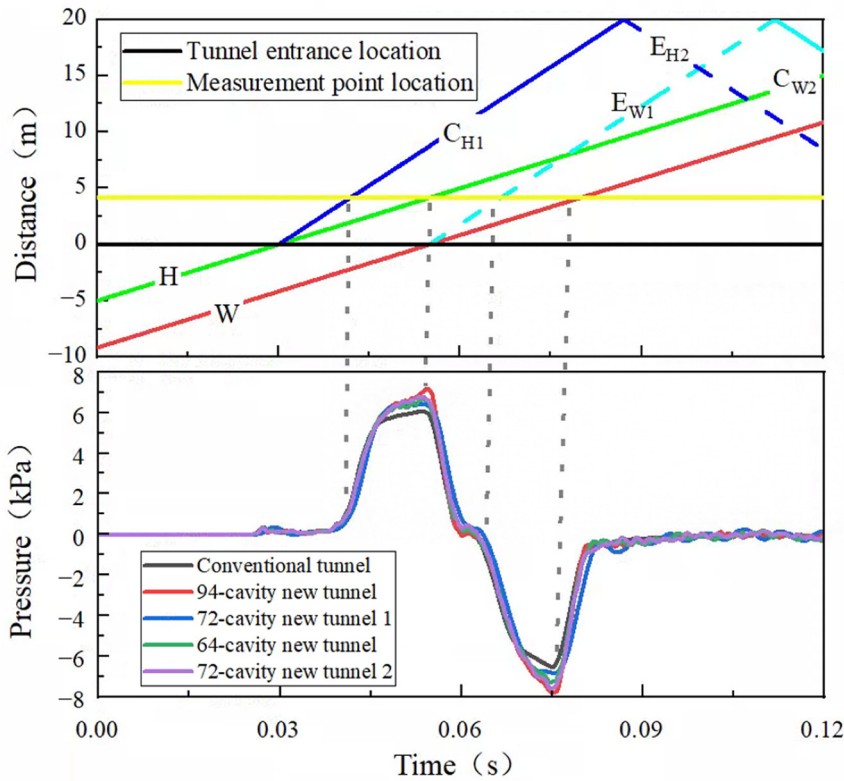

**Figure 14.** Comparison of wall pressures for different solutions.

### 3.3.3. Analysis of the Impact of Optimized Solutions on Micro-Pressure Waves

The data measured at measurement points M-1, M-2, and M-3 for the original scheme, 72-cavity new tunnel, and new 64-cavity tunnel were compared (see Table 5) and calculated as follows:

$$Y\% = \frac{P_{new} - P_{raw}}{P_{raw}} \times 100\% \tag{8}$$

where Y% denotes the mitigation rate for different tunnel scenarios, $P_{raw}$ denotes the data measured at the specified measurement points for the original tunnel scenario, and $P_{new}$ denotes the data measured at the specified measurement points for the new tunnel scenario.

**Table 5.** Comparison of the effects of different options for mitigating micro-pressure waves.

|  | **M-1** | **M-2** | **M-3** |
|---|---|---|---|
| Conventional tunnel | 1021.85 | 641.26 | 292.22 |
| 72-cavity new tunnel 1 remission rate | −27.1% | −29.3% | −30.7% |
| 64-cavity new tunnel remission rate | −25.6% | −27.8% | −27.4% |
| 72-cavity new tunnel 2 remission rate | −35.4% | −37.4% | −38.7% |

The two types of 72-cavity new tunnel schemes with Helmholtz resonators and 64-cavity new tunnel scheme can still significantly reduce the micro-pressure wave amplitude, with the mitigation rates of 72-cavity new tunnel 1 as 27.1%, 29.28%, and 30.65% at the three measurement points, respectively, and the mitigation rates fluctuating around 30%. The mitigation rates for the 64-cavity new tunnel were 25.6%, 27.8%, and 27.4%, with mitigation rates below 30%. According to the CEN European standard [30], a comparison of the amplitude data at 20 m and 50 m from the tunnel exit in Figure 15 shows that even if the same number of Helmholtz resonators are placed, the different arrangements will have an impact on the mitigation effect, indicating that different placements of Helmholtz resonators can lead to different mitigation rates of micro-pressure waves.

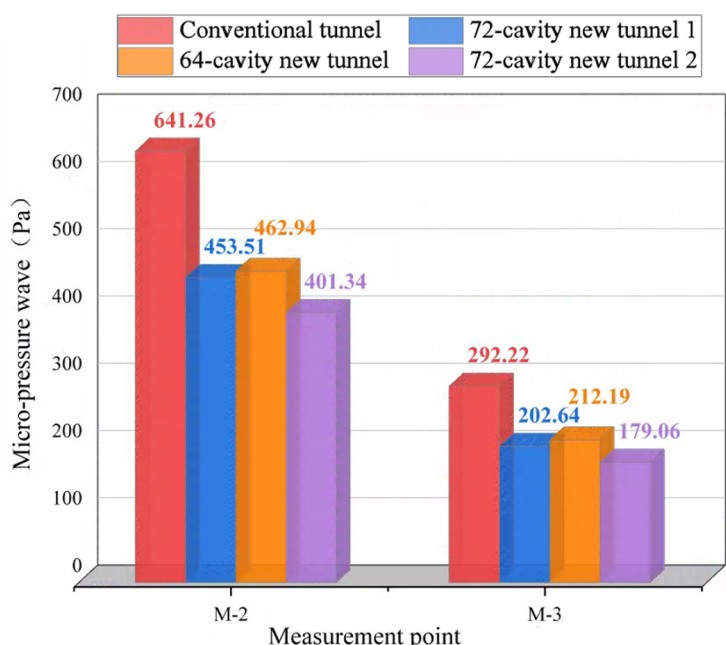

**Figure 15.** Effect of different layout options on micro-pressure waves.

The combined wall pressure rise rate and micro-pressure wave relief rate show that the 72-cavity new tunnel 1 arrangement can ensure the micro-pressure wave relief effect while controlling the increase in tunnel wall pressure to the maximum extent. This satisfies the requirements of the project. The fitted relationship between the tunnel exit distance and micro-pressure wave amplitude for the original tunnel and 72-cavity new tunnel 1 is shown in Figure 16, and the fitted equations and correlation coefficients are listed in Table 6. In this study, the amplitude of the micro-pressure wave at the exit of the tunnel is approximately inversely proportional to the 0.78th power of the distance from the tunnel exit.

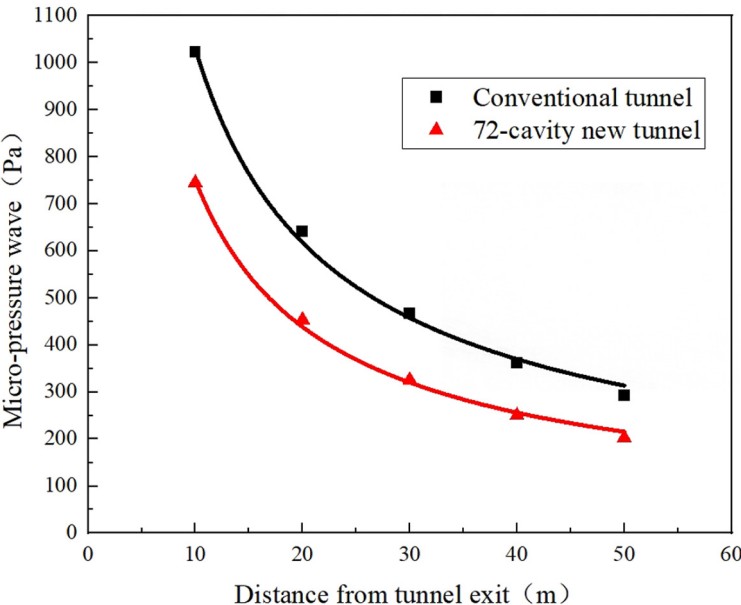

**Figure 16.** Micro-pressure wave fitting curves for different tunnel structures.

**Table 6.** Micro-pressure wave fitting curves for different tunnel structures.

| Tunnel Structure | Fitting Curves | Correlation Coefficient /$R^2$ |
|---|---|---|
| Conventional tunnel | y $=5657.39x^{-0.74}$ | 0.996 |
| 72-cavity new tunnel 1 | y $=4471.37x^{-0.78}$ | 0.997 |

## 4. Conclusions

In this study, the DDES method based on SST K–$\omega$ turbulence model and overlapping mesh technique was used to numerically simulate the aerodynamic characteristics of a high-speed maglev train passing through a tunnel. The main analysis involved examining the effect of Helmholtz resonators on the pressure amplitude at the tunnel wall and change in the micro-pressure wave amplitude at the tunnel entrance during the operation of high-speed maglev trains. Specifically, the correlation between the increase in the wall pressure amplitude, decrease in the micro-pressure wave amplitude, and change in the Helmholtz resonator arrangement was emphasized.

Based on the research in this study, the following main conclusions were obtained:

1. In this study, the initial compressional wave frequency generated by the high-speed maglev train passing through the tunnel corresponded to 15 Hz in the main frequency. Furthermore, the resonant frequency of the Helmholtz resonator was 14.63 Hz, and the upper and lower limits of the resonant band were 13.76 Hz and 16.13 Hz, respectively.

2. The Helmholtz resonator effectively reduced the micro-pressure wave amplitude at the tunnel exit by reducing the initial compressional gradient. In the conventional tunnel, the micro-pressure wave amplitude at 20 m and 50 m from the tunnel exit was 641.26 Pa and 292.22 Pa, respectively. In the 94-cavity new tunnel with the Helmholtz resonator, the micro-pressure wave amplitude at 20 m and 50 m from the tunnel exit was 436.91 Pa and 195.12 Pa, respectively. The Helmholtz resonator provided 31.87% and 33.23% relief of the micro-pressure wave amplitude at the two locations. However, the presence of the Helmholtz resonator increased the tunnel blockage ratio by 18.9%, resulting in an 18.62% increase in the tunnel wall pressure.

3. The 72-cavity new tunnel 1 is an optimized solution for the 94-cavity new tunnel, which exhibited 29.28% and 30.65% relief of micro-pressure waves at 20-m and 50-m measurement points at the tunnel exit, respectively, and only a 4.36% increase in tunnel wall pressure. The 72-cavity new tunnel 1 is an optimized solution for the 94-cavity new tunnel and ensured that the micro-pressure waves are relieved while maintaining an increase in the train surface pressure within acceptable limits and in line with engineering requirements.

4. In the 72-cavity new tunnel 1 scheme, the micro-pressure wave amplitude at the exit of the tunnel was approximately inversely proportional to the 0.78th power of the distance from the exit end of the tunnel, which approximates the fitted curve of the original tunnel scheme. This indicates that the presence of the Helmholtz resonator only reduces the initial compressional wave gradient and micro-pressure wave amplitude, and it does not affect the formation mechanism.

**Author Contributions:** Conceptualization, M.-Z.Y. and D.-Q.L.; Methodology, T.-T.L.; Formal analysis, P.Y. and D.-Q.L.; Resources, M.-Z.Y.; Writing—original draft preparation, D.-Q.L.; Writing—review and editing, D.-Q.L. and S.Z.; Supervision, S.Z.; Funding acquisition, M.-Z.Y. All authors have read and agreed to the published version of the manuscript.

**Funding:** The research was supported by the Graduate School-Enterprise Cooperation Project of Central South University, grant number 2022XQLH066.

**Institutional Review Board Statement:** Not applicable.

**Informed Consent Statement:** Not applicable.

**Data Availability Statement:** Data is contained within the article.

**Acknowledgments:** The authors acknowledge the China CRRC Science and Technology Research and Development Program (approval no. 2021CCA074), the National Numerical Wind Tunnel Project, and the High-Performance Computing Center of Central South University for providing computing resources.

**Conflicts of Interest:** The authors declare no conflict of interest.

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
