# Peer review of "Mitigation Effect of Helmholtz Resonator on the Micro-Pressure Wave Amplitude of a 600-km/h Maglev Train Tunnel"

_applsci, doi:10.3390/app13053124_

Round 1

Reviewer 1 Report

In this paper, the influence mechanism and reduction effect of Helmholtz resonators on the micro-pressure wave amplitude in the tunnel of a 600 km/h Maglev train are investigated by using the overlapping grid method, and their arrangement is optimally designed. The research content is relatively new, the research ideas and methods are scientific enough, and the overall flow is relatively smooth, which is an excellent paper However, there are some minor flaws in the article, and suggestions for improvement are as follows

1. The text of the abstract section needs to be a little more concise.

2. As you mentioned in your article, Helmholtz resonators were found to be effective in reducing the micro-pressure waves amplitude as early as 2017, so why don't you see other scholars' research on the application of Helmholtz resonators in this area in your introduction?

3. Heading 3.3.2, there is a problem of inaccurate language expression affecting the contextual coherence, and should be revised.

4. Lines 329-331, "Owing to ......speed" is not standardized and too colloquial

5. Lines 588-592, "According to ......waves" Although I can generally understand your meaning, but considering the reader's reading fluency, you should revise this sentence to express

The reviewers concluded that the current status of this paper meets the requirements for publication in this journal and it is recommended to accept it after completion of revisions.

Author Response

Thank you for your comments on my thesis. I have revised each of your comments, and here is my response:

Point 1:The text of the abstract section needs to be a little more concise.

Response 1:

Based on your suggestion, I have revised the abstract section as follows:

A 600-km/h maglev train can effectively close the speed gap between civil aviation and rail-based trains, thereby, alleviating the conflict between the existing demand and actual capacity . However, the hazards caused by the micro-pressure wave amplitude of the tunnel that occurs when the train is running at higher speeds are also unacceptable. At this stage mitigation measures to control the amplitude of micro-pressure waves generated by Maglev trains at 600 km/h within reasonable limits are urgent to develop new mitigation measures . In this study, a three-dimensional, compressible, unsteady SST K–ω equation turbulence model, and an overlapping grid technique were used to investigate the mechanism and mitigation effect of Helmholtz resonators with different arrangement schemes on the micro-pressure wave amplitude at a tunnel exit in conjunction with a 600-km/h maglev train dynamic model test. The results of the study showed that a pressure wave forms when the train enters the tunnel and passes through the Helmholtz resonator. This in turn leads to resonance of air column at its neck, which causes pressure wave energy dissipation as the incident wave frequency is in the resonator band. This suppresses the rise of the initial compressional wave gradient, resulting in an effective reduction in the micro-pressure wave amplitude at the tunnel exit. Com-pared to conventional tunnels, the Helmholtz resonator scheme with a 94-cavity 2 tunnel resulted in a 31.87% reduction in the micro-pressure wave amplitude at 20 m from the tunnel exit but a 16.69% increase in the maximum pressure at the tunnel wall. After the Helmholtz resonators were ar-ranged according to the 72-cavity optimization scheme, the maximum pressure at the tunnel wall decreased by 10.57% when compared with that before optimization. However, the micro-pressure wave mitigation effect at 20 m from the tunnel exit did not significantly differ from that before the optimization.

Point 2:As you mentioned in your article, Helmholtz resonators were found to be effective in reducing the micro-pressure waves amplitude as early as 2017, so why don't you see other scholars' research on the application of Helmholtz resonators in this area in your introduction?

Response 2:

In 2017, scholars Tebbutt et al. found that the Helmholtz resonator can effectively reduce the amplitude of micro-pressure waves when high-speed trains crossing tunnels at speeds below 400 km/h. However, since other mitigation measures for micro-pressure wave amplitude have already had mature ideas in the research direction, and considerable research results have been achieved, there is no need for research for the time being, both in terms of research cost and research value, so no scholars have conducted in-depth research on them yet . However, with the birth of the 600 km/h Maglev train, it was found that the existing mitigation measures could not control the micro-pressure wave amplitude within a reasonable range, and the development of a new buffer structure was imminent, so the study of the effect of Helmholtz resonators on micro-pressure waves only regained attention.

Point 3:Heading 3.3.2, there is a problem of inaccurate language expression affecting the contextual coherence, and should be revised.

Response 3:

I have reworked the title to strengthen the contextual coherence and the title is :Analysis of micro-pressure wave amplitude reduction effect

Point 4:Lines 329-331, "Owing to ......speed" is not standardized and too colloquial

Response 4:

Based on your suggestion, I have revised this sentence to : “ Due to the rapid changes in test pressure of a moving model locomotive, it is necessary to ensure that the sensing and measurement channels have sufficient response speed.”

Point 5:Lines 588-592, "According to ......waves" Although I can generally understand your meaning, but considering the reader's reading fluency, you should revise this sentence to express

Response 5:

Based on your suggestion, I have revised this sentence to :According to the CEN European standard [30], the comparison of amplitude data at 20 meters and 50 meters from the tunnel exit in FIG. 15 shows that even if the same number of Helmholtz resonators are placed, different arrangement modes will have an impact on the mitigation effect, indicating that different placement modes of Helmholtz resonators will lead to different relief rates of micro-pressure waves.

Reviewer 2 Report

Dear Authors, thanks you for your detailed manuscript and submission. Based on submitted manuscript, I would like to accept it with minor revision because of importance of completed work, but it may require certain improvements with below suggestions:
Introduction:
1) it is better to provide fundamental definition of micro-pressure waves before its introduction at L51.
2) Through-out the text, you provided reference number at the end of sentence, as a reader, there is nothing wrong, it is my opinion to use reference number along with author's name e.g. Yamanato et al. [4] in L65
Numerical Method:
1) It may be better to show schematic of Helmholtz Resonator in the study context along with mentioning different design features (neck length, cavity sizing etc).
2) Figure 3, should have arrows then curved lines to direct freestream condition.
Results and Discussion,
The results are discussed based on few point measurements, the discussion has been a bit superficial. With the 3-D simulation data, I would like to see authors to represent and analyze their rich results in further details. Few recommendation: 
1) Comparing the pressure data in x-T (location/time) at the centerline/wall of tunnel. (use examples of shock-tunnels) with conventional / tunnels with controls. 
2) Fig. 10, what is sampling rate here?
3) Not only the amplitude is important, You should mention, how the energy is distributed in frequency spectrum with different cavity controls.

I hope authors will revise the discussion accordingly.
Regards,

Author Response

Thank you for your comments on my thesis. I have revised each of your comments, and here is my response:

Introduction:

Point 1: it is better to provide fundamental definition of micro-pressure waves before its introduction at L51

Response 1:

Before L51, I added the definition of micro-pressure waves as follows:

When a high-speed maglev train passes through a tunnel, an initial compression wave is formed in front of the vehicle, which propagates forward at the local speed of sound and reaches the exit of the tunnel, and part of it radiates outward in the form of a low frequency pulse wave, forming a micro-pressure wave.qa1

Point 2: Through-out the text, you provided reference number at the end of sentence, as a reader, there is nothing wrong, it is my opinion to use reference number along with author's name e.g. Yamanato et al. [4] in L65

Response 2:

Thank you for your guidance. I agree with you very much and have revised all the contents of the introduction according to your guidance.

Numerical Method:

Point 1: It may be better to show schematic of Helmholtz Resonator in the study context along with mentioning different design features (neck length, cavity sizing etc).

Response 1:

According to your suggestion, I modified the structure schematic in the paper, added markups, and showed the design details in more detail.

Point 2:Figure 3, should have arrows then curved lines to direct freestream condition.

Response 2:

Figure 3 has been modified according to the requirements

Results and Discussion:

Point 1:Comparing the pressure data in x-T (location/time) at the centerline/wall of tunnel. (use examples of shock-tunnels) with conventional / tunnels with controls. 

Response 1:

May be due to my expression problem, so that you have a misunderstanding at the time of review, here I express my deep apology to you.

In Section 3.1 and Section 3.3.1 of this paper, I proved that there is only a small difference in the data measured by different wall measuring points on the cross section of the tunnel in this case. The wall pressure of different tunnel schemes is discussed. The specific comparison is as follows.

I have revised the expression to your satisfaction.

Point 2:Fig. 10, what is sampling rate here?

Response 2:

The sampling rate is 943.

Point 3: Not only the amplitude is important, You should mention, how the energy is distributed in frequency spectrum with different cavity controls

Response 3:

Thank you for your advice and inspiration, which has given me great help.

In this paper, I focused on studying the influence mechanism of Helmholtz resonator on micro-pressure waves, and the influence of distribution mode on tunnel wall pressure and micro-pressure wave amplitude. As you pointed out, my next research direction is on the influence mechanism of the size of Helmholtz resonator (volume, cavity length, hole area, etc.) on the amplitude of micro-pressure wave. These problems are more complicated and require me to further study.

Thank you again for your guidance.

Reviewer 3 Report

Micro pressure wave for a 600-km/h maglev train tunnel is alleviated using Helmholtz resonator. Comments are as below.

1.  line 140: The statement "maintain the surface layer y+ within a reasonable range," should be more specific.

2. "H" is denoted as the train height (line 142) and train head displacement (line 378).

3. The authors may give more details for Helmholtz resonator arrangements (lines 168-179), e.g. cavities grouped in two or three.

4. Fig. 2b: The scale is not readable.

5. The instrumentation for pressure measurements is not specified.

6. line 275: free flow ?

Author Response

Thank you for your comments on my thesis. I have revised each of your comments, and here is my response:

Point 1: line 140: The statement "maintain the surface layer y+ within a reasonable range," should be more specific.

Response 1:

According to your suggestion, after careful consideration, the content will be adjusted as follows :The train model was scaled down to 1:20 prior to the numerical simulation because the full y+ wall treatment was used and the y+ needed to be controlled to be less than 1 or greater than 30.

Point 2: "H" is denoted as the train height (line 142) and train head displacement (line 378).

Response 2:

Revised to "h" for train height and "H" for train head displacement, and corrections have been made to the full paper.

Point 3: The authors may give more details for Helmholtz resonator arrangements (lines 168-179), e.g. cavities grouped in two or three.

Response 3:

Based on your suggestion, I added the description of the program to make it more detailed, and the revised result is as follows:

  1. Conventional tunnel without Helmholtz resonators and without any additional structures.
  2. A 94-cavity new tunnel comprising 94 cavities arranged continuously without in-tervals on both sides of the tunnel, starting at 4.78h from the entrance and ending at 6.83h from the exit of the tunnel.
  3. A 72-cavity new tunnel 1, the 72 cavities were divided into 24 groups, each group consisting of 3 cavities, with an interval of 1.76h between each group, starting inside the tunnel at 4.78h from the entrance and ending at 6.83h from the tunnel exit.
  4. A 64-cavity new tunnel, the 64 cavities were divided into 32 groups, each group consisting of 2 cavities, with an interval of 1.76h between each group, starting inside the tunnel at 2.39h from the entrance and ending at 6.83h from the tunnel exit.

5.A 72-cavity new tunnel 2, the 72 cavities were divided into 24 groups, each group consisting of 3 cavities, with an interval of 1h between each group, arranged from 2.39h inside the tunnel from the entrance and ending at 15.18h from the exit of the tunnel. The specific programmer is listed in Table 1.

Point 4:Fig. 2b: The scale is not readable.

Response 4:

Changed the annotation of the figure, adding units to it to increase readability

Point 5: The instrumentation for pressure measurements is not specified.

Response 5:

Section 2.4.2 was modified to include the following detailed description of pressure sensors in the paper.

The entire range output was fed into a high-speed A/D converter for acquisition as a standard voltage signal of 0–5 V.  Each pressure signal channel has an independent circuit structure and A/D converter, and synchronous sampling is controlled by the same time-based signal, thereby ensuring fast data acquisition and consistency in time and space for each channel.

Point 6: line 275: free flow ?

Response 6:

After careful proofreading, the name was changed to "free-stream", equivalent to the pressure-far-field boundary condition
